# Synergetic electrode architecture for efficient graphene-based flexible organic light-emitting diodes

Jaeho Lee[1,2,*], Tae-Hee Han[3,*], Min-Ho Park[3,*], Dae Yool Jung[1,2], Jeongmin Seo[4], Hong-Kyu Seo[3], Hyunsu Cho[1,2,†], Eunhye Kim[1,2], Jin Chung[1,2], Sung-Yool Choi[1,2], Taek-Soo Kim[4], Tae-Woo Lee[3] & Seunghyup Yoo[1,2]

Graphene-based organic light-emitting diodes (OLEDs) have recently emerged as a key element essential in next-generation displays and lighting, mainly due to their promise for highly flexible light sources. However, their efficiency has been, at best, similar to that of conventional, indium tin oxide-based counterparts. We here propose an ideal electrode structure based on a synergetic interplay of high-index $TiO_2$ layers and low-index hole-injection layers sandwiching graphene electrodes, which results in an ideal situation where enhancement by cavity resonance is maximized yet loss to surface plasmon polariton is mitigated. The proposed approach leads to OLEDs exhibiting ultrahigh external quantum efficiency of 40.8 and 62.1% (64.7 and 103% with a half-ball lens) for single- and multi-junction devices, respectively. The OLEDs made on plastics with those electrodes are repeatedly bendable at a radius of 2.3 mm, partly due to the $TiO_2$ layers withstanding flexural strain up to 4% via crack-deflection toughening.

[1] School of Electrical Engineering, Korea Advanced Institute of Science and Technology (KAIST), Daejeon 305-701, Republic of Korea. [2] Graphene Research Center, KI for Nanocentury, KAIST, Daejeon 305-701, Republic of Korea. [3] Department of Materials Science and Engineering, Pohang University of Science and Technology (POSTECH), Pohang, Gyungbuk 790-784, Republic of Korea. [4] Department of Mechanical Engineering, KAIST, Daejeon 305-701, Republic of Korea. * These authors contributed equally to this work. † Present address: Electronics and Telecommunications Research Institute (ETRI), Daejeon 305-700, Republic of Korea. Correspondence and requests for materials should be addressed to T.-W.L. (email: twlee@postech.ac.kr) or to S.Y. (email: syoo.ee@kaist.edu).

Form-factor advantages such as flexibility and stretchability have brought organic light-emitting diodes (OLEDs) great attention for use in emerging devices like wearable, roll-type or foldable displays that call for mobility, deformability and/or expandability[1,2]. Full realization of such potential, however, is non-trivial and requires significant effort in various aspects such as development of low-temperature thin-film transistor technologies and/or flexible encapsulation[3]. Another important aspect to consider for highly flexible displays is to find transparent electrodes (TEs) that can replace indium tin oxide (ITO) electrodes, which typically suffer from limited flexibility and potential cost issues associated with high demand and/or unstable supply[4]. In this regard, several kinds of flexible transparent electrodes such as metal nanowires, carbon-based materials (for example, carbon nanotubes or graphene) and metal grids coupled with other types of TEs were proposed for OLEDs[4–9]. Among them, graphene—an atomically thin sheet of $sp^2$-hybridized carbon atoms—has been regarded as promising because the unique combination of its inherent thinness and superior electrical characteristics is expected to allow for ideal transparent electrodes that have low or no optical haze, smooth surface topology preventing roughness-induced electrical short and degradation, high transmittance, low sheet resistance and a high degree of flexibility all at the same time[10,11]. Facile preparation of high-quality graphene on large-scale plastic substrates has recently been demonstrated for transparent electrodes with mass-production-compatible methods by industrial sectors, demonstrating their practical viability[11,12]. Both single-layer graphene (SLG) and multi-layer graphene (MLG) were shown to have performance adequate for TEs in OLEDs[13,14]. In particular, efficient graphene-based OLEDs were achieved with the help of hole injection layers (HILs) used to overcome their relatively low work function, which is known to be around 4.5 eV. Such injection layers include conducting polymers with gradient electronic properties called a self-organized gradient HIL (GraHIL)[13], and thin $MoO_3$ buffer layers coupled with conducting polymers of poly (3,4-ethylenedioxythiophene): polystyrene sulfonate (PEDOT: PSS)[14]. Nevertheless, the power efficiency and external quantum efficiency (EQE) of these state-of-the-art graphene-based OLEDs were still on a par with those of ITO-based OLEDs, unless a bulky hemispherical lens was used for outcoupling enhancement[14]. Securing the highest possible efficiency is particularly important for flexible displays applied to highly portable or wearable applications, as they often have to rely on batteries with a fairly limited energy capacity due to constraints in size, weight or form factors. This calls for the development of a device architecture that can maximize the efficiency of graphene-based OLEDs. A key challenge is to develop a structure or methodology to unlock their full optical potential yet retains graphene's merits in form factors as much as possible.

To this end, we here explore a simple electrode architecture based on high-index $TiO_2$ layers and low-index HILs sandwiching graphene electrodes. With the optical design that takes a full advantage of the synergetic collaboration between the high- and low-index layers controlling both cavity resonance enhancement and loss to surface plasmon polariton (SPP), the proposed graphene-based OLEDs exhibit ultrahigh EQE that is unprecedented in those using graphene as a transparent electrode. Furthermore, unusually high resistance of $TiO_2$ to flexural strain is revealed that enables plastic OLEDs that are not only efficient but also highly flexible.

## Results

### Optical design of high-efficiency graphene-based OLEDs. The efficiency of OLEDs is limited ultimately by a finite outcoupling

efficiency that is typically around 20% at best for devices with isotropic emitters. Among various outcoupling enhancing schemes proposed to date[15], a method based on microcavity resonance has advantages in that it maintains a planar geometry without using any micro/nano structuring or lenticular structure so that the OLEDs made thereof are not subject to electrical shorts and optical artefacts unfavourable for display application (for example, blurring, diffraction and so on) yet can exhibit highly pure, saturated red (R), green (G), blue (B) primary colours with the enhanced efficiency[16,17]. With a weak but still significant microcavity effect, ITO-based OLEDs can also be optimized for maximum efficiency, without incurring any optical haze or blurring, simply by adjusting the thickness of ITO layers; in this scheme, first-order, $3\lambda/4$ cavity design, where $\lambda$ refers to wavelength, is achieved with the open end of the cavity mode placed at the ITO/substrate interface[18,19]. In the case of ITO-free, graphene-based OLEDs, however, graphene electrodes cannot be made thick enough to control the cavity length, which makes it challenging to enhance efficiency by the resonance effect[20].

As an alternative measure, one may place a thin layer with high refractive index ($n_H$) underneath graphene layers as shown in Fig. 1a in a way similar to dielectric capped thin-metal electrodes[17,20–22]. Because light can pass through the graphene with little phase difference due to the inherent thinness of a graphene electrode, $3\lambda/4$ cavity design can be achieved conveniently, for example, with the optical thickness of the organic plus injection layers and the high refractive index layer set at approximately $2\lambda/4$ and $\lambda/4$, respectively (Fig. 1b). In particular, the '$\lambda/4$-thick' high-index layer functions also as a metal-free, dielectric mirror that enables significant reflectance from bottom electrode assembly ($R_{bot}$) for light incident from organic layers, as can be confirmed in the graphs shown at the bottom of Fig. 1c,d. This provides graphene-based OLEDs with opportunities to enhance their efficiency via the resonance effect, which would not be possible otherwise. It is shown that $R_{bot}$ and concomitant resonance enhancement increase with $n_H$ (Supplementary Fig. 1). The high refractive index is also beneficial in terms of flexibility because the target optical thickness ($n_H d_H$) can be achieved with low physical thickness $d_H$, and because bending-induced crack of a given film generally forms at a lower onset strain with a larger physical thickness[23]. A successful high-index layer in this scheme should thus have as large $n_H$ as possible. Furthermore, it should be transparent, and, most of all, should not be damaged by a given graphene transfer process. Among various candidates, a sputtered $TiO_2$ layer is transparent in a visible spectral range and has a relatively high refractive index ($n = 2.5$). Unlike other high-index layers we tried, the $TiO_2$ layer turns out to have chemical endurance robust against graphene transfer processes that use solvent, such as acetone or isopropyl alcohol (IPA), allowing for formation of a high-quality graphene electrode on its top (Supplementary Fig. 2).

It is noteworthy that a higher order micro-cavity structure (for example, second order, $5\lambda/4$ cavity) based on a thick organic stack may also be possible. In this work, however, a first-order cavity design was first chosen because it results in a stronger Purcell effect and fewer waveguide modes than the higher-order design, ultimately leading to larger outcoupling efficiency and EQE[24,25]. Challenges still remain for the first-order cavity design because the ideal location of an emission zone is limited at the first antinode from the organic/metal interface, making it difficult to reduce SPP modes through longer emitter-to-metal distance[25,26]. An alternative method typically used for SPP reduction is to incorporate internal wavy structures or corrugations[27], but tolerance to electrical shorts can often be compromised by such approaches. The method we adopt here is to include a layer with a low refractive index, which has been

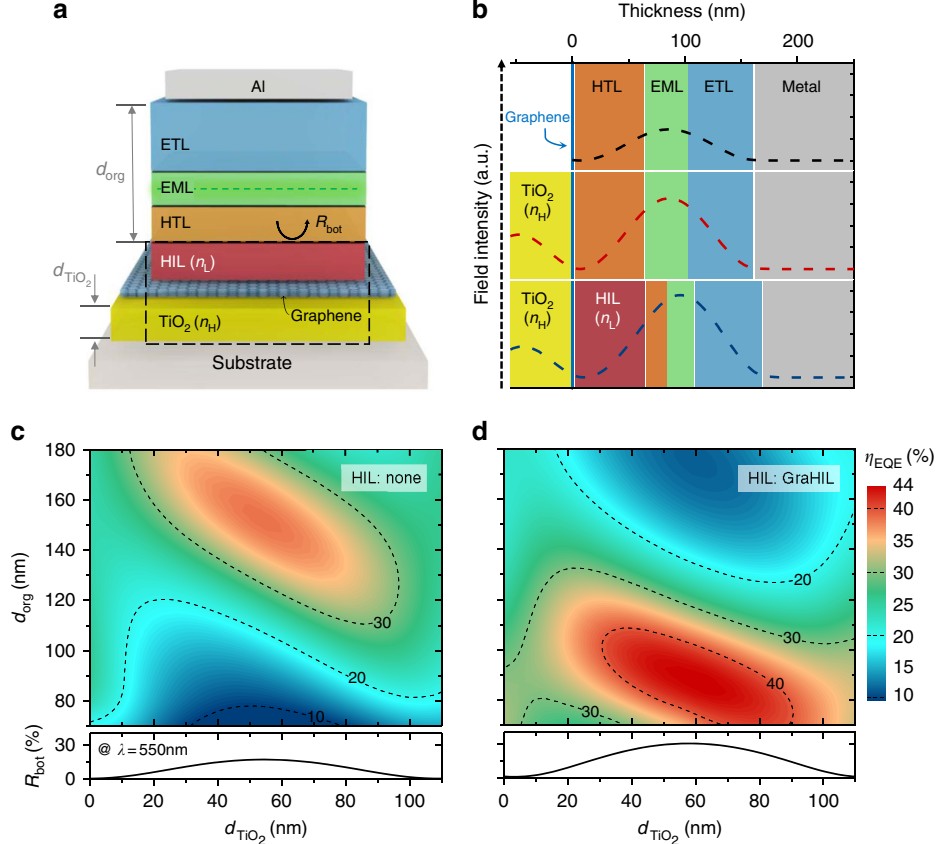

**Figure 1 | Design overview of proposed high-efficiency graphene-based OLEDs.** (**a**) Schematic device structure of the proposed OLEDs. (**b**) Electromagnetic field intensity distribution (dashed lines) of the OLEDs under study for their respective first-order cavity design. The field-intensity distribution of the graphene-based OLED without TiO$_2$ is also shown for comparison for the case where the thickness values of organic layers are same as the OLEDs with TiO$_2$ but without low-index HILs. (**c,d**) Calculated maximum external quantum efficiency ($\eta_{EQE}$) of graphene-based OLEDs with TiO$_2$ under-layer as a function of TiO$_2$ and organic layer thickness ($d_{TiO2}$ and $d_{org}$, respectively). Dashed lines represent contour lines for $\eta_{EQE}$ of 10, 20, 30 or 40% as indicated in the graph. (**c**) Without a low-index HIL; (**d**) with GraHIL as a low-index HIL. On the bottom of each case, the reflectance ($R_{bot}$) from the bottom electrode assembly (TiO$_2$/graphene/(HIL)) as a whole is presented as a function of $d_{TiO2}$ for light incident from the organic layers.

shown to effectively reduce SPP modes[28,29]. Fortunately, GraHIL or a self-organized gradient hole-injection layer composed of PEDOT:PSS and tetra-fluoroethylene-perfluoro-3, 6-dioxa-4-methyl-7-octenesulphonic acid copolymer (PFI), which was previously proposed by authors and shown to be very effective in improving hole injection in graphene-based OLEDs[13], has a refractive index around 1.42 at $\lambda$ of 550 nm (Supplementary Fig. 3), which is much lower than those of typical organic layers ($n = 1.8$). With the synergetic interplay of high- and low-index layers sandwiching graphene layers, enhancement of the cavity-resonant effect and reduction of SPP modes may thus be done simultaneously so that the EQE can be significantly boosted even without outcoupling structures.

Figure 2a,b present the optical simulation results based on an advanced classical electromagnetic theory summarized by Furno *et al.*[19] over the whole visible spectral range, and at $\lambda$ of 550 nm. The formalism takes into account Purcell factor, dipole orientation effect, and excitations to SPP and waveguide modes. A comprehensive inclusion of all these factors turns out critical for a precise analysis and quantitative design as under- or overestimation can occur when one uses a simplified approach that do not account for one or more of those effects (see Supplementary Fig. 4; Supplementary Table 1 and Supplementary Note 1 for examples). The results indicate that the use of GraHIL redistributes the relative power contents among various modes towards smaller in-plane wave vectors

such that the power coupled to SPP modes is reduced and the outcoupled portions are further enhanced (Supplementary Fig. 5a). It can also be seen in Supplementary Table 2 that use of the TiO$_2$ layer underneath graphene effectively suppresses the amount of the power coupled to waveguided and substrate modes via resonance enhancement. This leads to a significant increase in outcoupling efficiency, although loss to evanescent modes is also shown to increase due to the shift of transverse-electric waveguided modes towards higher normalized in-plane wave vectors (Supplementary Fig. 5b). The combined use of high-index (H) TiO$_2$ and low-index (L) GraHIL layers, results in an ideal situation where resonance enhancement is large, yet loss to SPP/evanescent modes is mitigated. In addition, the combined use creates a situation similar to HL-stacks, typically used for multilayer thin-film coating[30], and even further enhances $R_{bot}$ (Fig. 1d), allowing for additional enhancement in the Fabry–Perot resonance effect. What makes the proposed technology unique and advanced with respect to conventional technologies is indeed the synergetic collaboration of these high- and low-index layers that enable optical management of both resonance effect and SPP loss to the advantage of maximal outcoupling; and additional resonance enhancement via HL stacking. Together with their native compatibility with desired electrical properties (for example, efficient hole injection), all these optical benefits allow one to fully unlock even the hidden potential of what planar OLEDs can truly offer. Contour plots obtained for optically

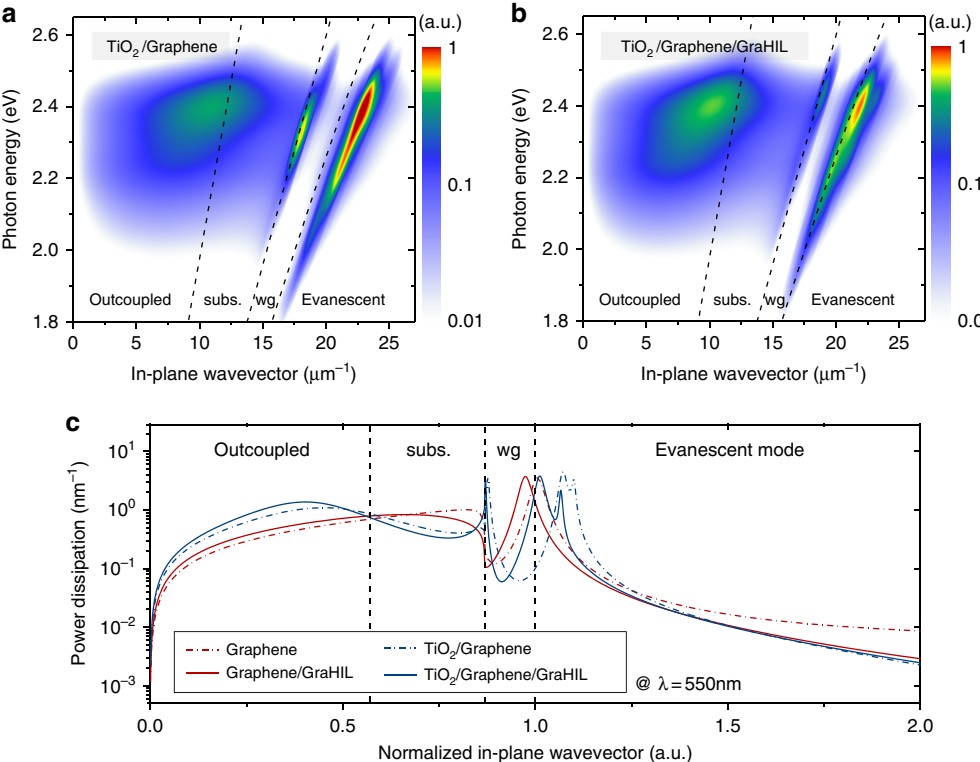

**Figure 2 | Synergetic optical effect of TiO$_2$ as a high-index layer and GraHIL as a low-index HIL. (a,b)** Calculated power dissipation spectra weighted with the emitter spectrum in arbitrary units (mapped as a colour defined in the colour bars.) versus in-plane wave vector: (**a**) with TiO$_2$ but without GraHIL; (**b**) with both TiO$_2$ and GraHIL. The black dashed lines indicate border lines dividing representative optical modes including outcoupled, substrate-confined (subs), waveguided (wg) and evanescent modes. (**c**) Calculated power dissipation versus normalized in-plane wave vector at $\lambda = 550$ nm for various electrode structures under study. For the devices with TiO$_2$, $d_{TiO2}$ was fixed at 55 nm and $d_{org}$ was chosen for optimal conditions in each case. For the devices without TiO$_2$, $d_{org}$ was set at the same value as that of each counterpart in the devices with TiO$_2$.

available maximum EQE ($\eta_{EQE}^{(max)}$) as a function of organic ($d_{org}$) and TiO$_2$ thickness ($d_{TiO2}$) presented in Fig. 1c,d show that $\eta_{EQE}^{(max)}$ can be as high as 44% in devices with both GraHIL and TiO$_2$ layers while it would be limited to 38% without GraHIL or to 31% without TiO$_2$. Optimal conditions do correspond to a situation where $d_H$ (with H being TiO$_2$) gets close to $\lambda/(4n_H)$ (55 nm for $\lambda = 550$ nm). It can also be noted that $\eta_{EQE}^{(max)}$ of 44% is higher than those expected for optimized, conventional thin-metal based cavity OLEDs with the same emitter. This can be attributed to the fact that the proposed architecture is less subject to the loss due to photon absorption and/or SPP mode excitation within semi-transparent electrodes (see Supplementary Fig. 6 and Supplementary Table 3 for comparison with those of conventional thin-metal based cavity OLEDs).

**Device performance of fabricated graphene-based OLEDs.** Inspired by the simulation results shown above, we fabricated green OLEDs based on phosphorescent emitters of bis(2-(2-pyridinyl-N)phenyl-C)(acetylacetonato) iridium (III) (Ir(ppy)$_2$ acac) in configuration of glass/anode/(HIL)/OS1/LiF/Al with different anode/(HIL) structures of TiO$_2$/graphene/GraHIL; graphene/GraHIL; and ITO (185 nm)/GraHIL, where OS1 refers to an organic multilayer stack defined in Methods. Experimental results presented in Fig. 3 show that the maximum EQE, power efficiency and current efficiency obtained for the TiO$_2$/graphene/GraHIL-based device are as high as 40.8%, 160.3 lm W$^{-1}$ and 168.4 cd A$^{-1}$, respectively, while those values are limited to 31.7%, 112.6 lm W$^{-1}$ and 119.0 cd A$^{-1}$ for the graphene/GraHIL-based device, and 27.4%, 104.3 lm W$^{-1}$ and 106.2 cd A$^{-1}$

for ITO/GraHIL-based device. The observed high EQE in the TiO$_2$/graphene/GraHIL-based device confirms the synergetic role of both TiO$_2$ and GraHIL layers, and is consistent with the simulation results. With a half-ball lens optically coupled to the back of a substrate, the TiO$_2$/graphene/GraHIL-based device exhibits EQE, power efficiency and current efficiency as high as 64.7%, 250.4 lm W$^{-1}$ and 257.0 cd A$^{-1}$, respectively, further illustrating the promising potential of the proposed graphene-based OLEDs. We then further extended the proposed scheme to a tandem, multi-junction OLED, in which the net EQE is improved (at the expense of voltage) because it is eventually given as the sum of those of its individual OLEDs[31]; this approach is popular as the improved EQE generally leads to longer operation lifespan[32]. TiO$_2$/graphene/GraHIL-based multi-junction OLEDs containing a charge-generation layer (CGL) of 2, 9-dimethyl-4, 7-diphenyl-1, 10-phenanthroline (BCP):Li/MoO$_3$ exhibit EQE and power efficiency as high as 62.1% and 120.8 lm W$^{-1}$ (103.2% and 183.5 lm W$^{-1}$ with the half-ball lens), demonstrating the state-of-the-art performance and the versatile applicability of the proposed electrode structure. The simulation indicates that the observed value corresponds to the sum of EQE values of equivalent second-order cavity structures (one with the emission zone near second antinode from the cathode; and the other with the emission zone near first antinode from the cathode) with the electrical balance factor of 0.89 (see Supplementary Fig. 7, Supplementary Table 4 and Supplementary Note 2 for the detailed structure for multi-junction OLEDs as well as the detailed discussion on the simulation results).

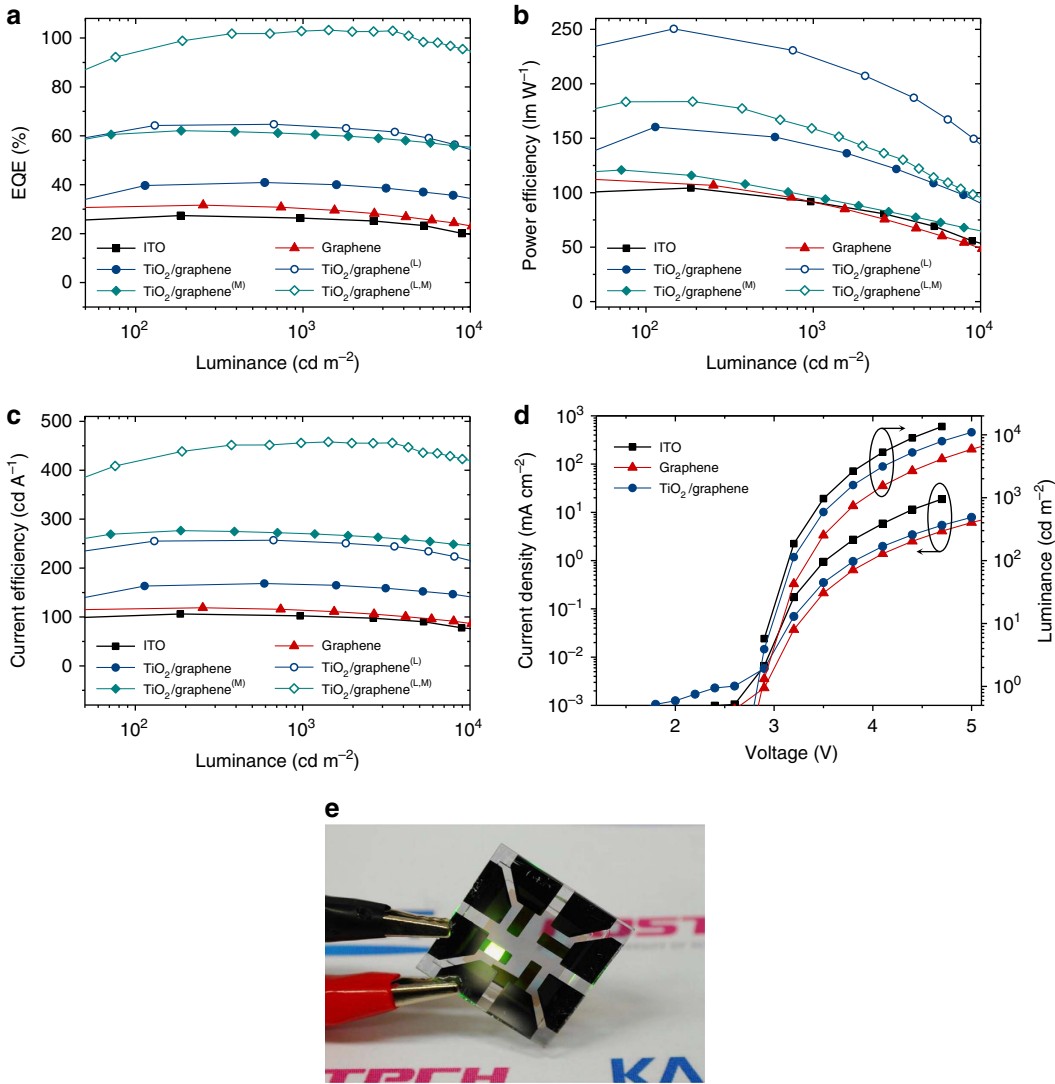

**Figure 3 | Device performance of graphene-based OLEDs under study.** (**a**–**c**) EQE (**a**) power efficiency (**b**) and current efficiency (**c**) versus luminance characteristics of graphene-based OLEDs with and without $TiO_2$ under-layers. The data for ITO-based control OLEDs are also shown for comparison. All the devices in this figure contain GraHIL as low-index hole-injection layers and are based on glass substrates. In (**a**–**c**) legends with (L), (M) and (L,M) correspond to the data obtained with a half-ball lens attached, with a multi-junction architecture, and with both a multi-junction architecture and a half-ball lens, respectively. (**d**) Current density ($J$)–voltage ($V$)–luminance ($L$) characteristics. The detailed structure and $J$–$V$–$L$ characteristics of a multi-junction device are represented in Supplementary Fig. 7. (**e**) The picture of an OLED with the proposed $TiO_2$/graphene/GraHIL electrode in operation. For the devices with $TiO_2$, $d_{TiO2}$ was fixed at 55 nm and $d_{org}$ was chosen as the optimal condition for maximum EQE. For the devices without $TiO_2$ and the ITO-based devices, $d_{org}$ was set at the same value as that of the corresponding devices with $TiO_2$.

**Mechanical properties of the proposed electrodes and OLEDs.** The success of the graphene-based OLEDs would not be complete unless the advantage of graphene electrodes in flexibility can be utilized to a significant degree. To test the suitability of the proposed structure for flexible OLEDs, OLED devices were made in configurations of $(TiO_2)$/graphene/PEDOT:PSS/OS2/LiF/Al on 50-µm-thick polyethylene terephthalate (PET) substrates (see Methods for the multilayer stack used for flexible OLEDs (OS2)). It is noteworthy that the commonly available PEDOT:PSS (AI4083 by Clevios; $n = 1.56$ at λ of 550 nm) can play essentially the same optical role as GraHIL, only with a little compromise in its optical effect due to its refractive index being slightly higher than that of GraHIL (see Supplementary Fig. 8 and Supplementary Table 5). As can be seen in Fig. 4, the proposed flexible OLEDs exhibit high EQE and power efficiency comparable to their glass-based counterparts (shown as black triangles in Fig. 4), demonstrating that the proposed structure

and its processes can be extended to plastic substrates without problems (see Table 1 for full summary of the performance of all the devices studied in this work). Furthermore, these devices are shown to remain intact and operate well even after 1,000 bending cycles at a radius of curvature ($r_C$) as small as 2.3 mm. This can be considered remarkable because the proposed OLEDs contains oxide layers, which are often regarded as brittle and thus prone to bending-induced fracture even at relatively low strain.

To understand better the origin of the observed level of flexibility in the proposed OLEDs, the mechanical property of $TiO_2$ layers deposited on PET substrates was characterized, based on a 3-point bending test (ASTM D790; static loading) as shown in Fig. 5a (see Supplementary Fig. 9 for further details on the test method). Scanning electron microscopy images of the tested samples reveal that 60-nm-thick $TiO_2$ can withstand flexural strain ($ε_f$) as large as 4% without developing bending-induced cracks, while ITO (200 nm) and indium zinc oxide (IZO;

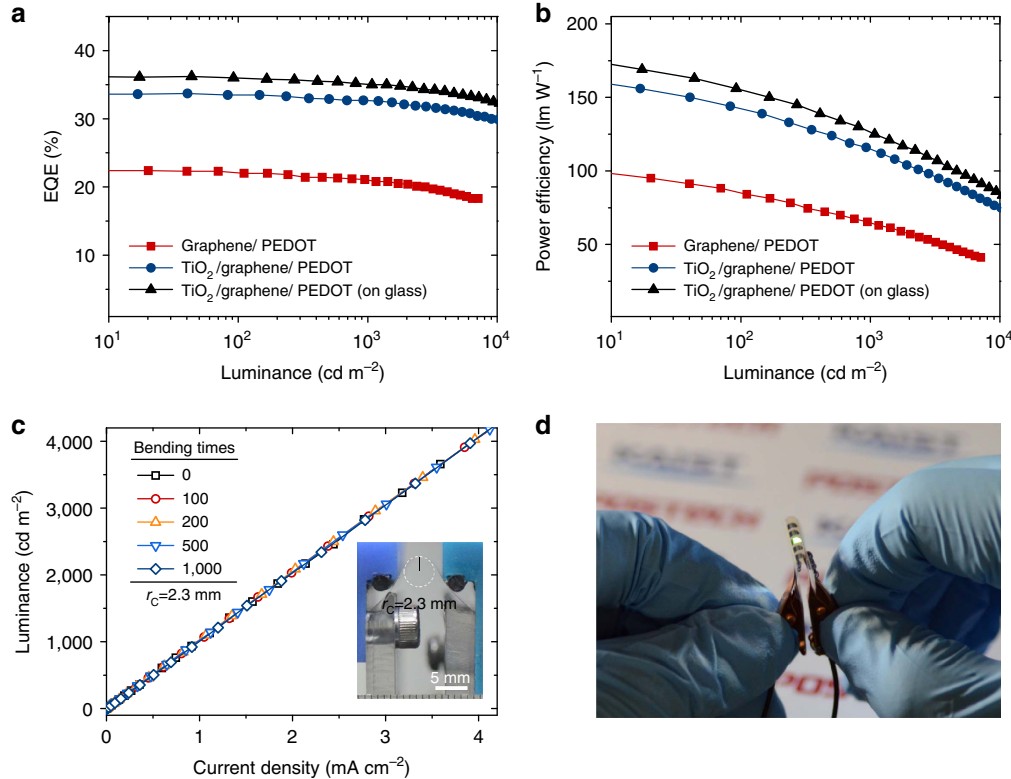

**Figure 4 | Performance of graphene-based flexible OLEDs fabricated on 50 μm PET substrate. (a)** EQE versus luminance characteristics of graphene-based flexible OLEDs with and without $TiO_2$ under-layers. Those of an OLED with $TiO_2$ on a glass substrate are also shown for comparison. For devices presented in this figure, 50-nm-thick PEDOT:PSS (AI4083) was used as a low-index HIL in all cases. **(b)** Power efficiency versus luminance characteristics. **(c)** Luminance versus current density characteristics measured after repeated bending at radius of curvature ($r_C$) of 2.3 mm. Inset: the photograph of an OLED bent at $r_C$ of 2.3 mm placed on a custom-made cyclic bending tester. The OLED is located on the top side of the substrate in the picture and thus is under tensile strain. **(d)** Photograph of the proposed flexible $TiO_2$/graphene OLEDs in operation.

**Table 1 | Performance summary of OLEDs studied in this work.**

| OLED device structure* | EQE (%) | | Power efficiency (lm W$^{-1}$) | |
|---|---|---|---|---|
| | Max. | @ 5,000 cd m$^{-2}$ | Max. | @ 5,000 cd m$^{-2}$ |
| Glass/TiO$_2$/graphene/GraHIL/OS1/LiF/Al | 40.8 (64.7$^†$) | 37.2 (59.8) | 160.3 (250.4) | 110.2 (178.7) |
| Glass/graphene/GraHIL/OS1/LiF/Al | 31.7 | 26.2 | 112.6 | 63.8 |
| Glass/ITO/GraHIL/OS1/LiF/Al | 27.4 | 23.5 | 104.3 | 70.4 |
| Glass/TiO$_2$/graphene/GraHIL/OS1′/BCP:Li/MoO$_3$/OS1′/LiF/Al$^‡$ | 62.1 (103.2) | 57.5 (98.8) | 120.8 (183.5) | 74.6 (115.8) |
| PET/TiO$_2$/graphene/PEDOT/OS2/LiF/Al | 33.8 | 31.1 | 155.8 | 88.0 |
| PET/graphene/PEDOT/OS2/LiF/Al | 22.4 | 18.9 | 95.1 | 45.5 |
| Glass/TiO$_2$/graphene/PEDOT/OS2/LiF/Al | 36.2 | 33.6 | 168.9 | 98.6 |

*The detailed multilayer stacks for OS1 and OS2 are presented in Methods.
$^†$Shown in ( ) are the values obtained with a half-ball lens attached to the back of the substrate.
$^‡$The detailed stack structure for organic layer (OS1′) in multi-junction devices are presented in Supplementary Fig. 7.

60 nm) electrodes withstand $\varepsilon_f$ of 1% but start to exhibit cracks at 2% (Fig. 5b). It is noteworthy that the crack developed in the $TiO_2$ sample with $\varepsilon_f$ of 5% propagates in a zig-zag fashion, presenting a clear contrast to the cracks developed in the ITO and IZO samples, which appear as almost straight lines. This indicates that the observed significant crack resistance of $TiO_2$ under relatively large strain can be attributed to 'crack deflection toughening' mechanism, which is known to be effective in relieving locally high stresses at cracks[33]. As shown in Fig. 5c,d, consistent results are also observed in electrical resistance measurements, which provide macroscopic information on change in the integrity of films after repeated bending (dynamic loading). While the sheet resistance of 60-nm-thick IZO exhibits a significant increase after

only 10 cycles at strain of ca. 2%, that of the $TiO_2$/graphene electrodes remains almost unchanged even after 10,000 bending cycles at strain of 4%, explaining the origin of the observed high degree of flexibility of the OLEDs based on $TiO_2$/graphene electrodes.

## Discussion

One may note that the proposed technology works in principle for a specific target wavelength and may accompany unwanted side effects such as angular colour shift, as it is based on resonance phenomena. Nevertheless, it can be shown that it is applicable to all red (R), green (G) and blue (B) pixels simultaneously with a

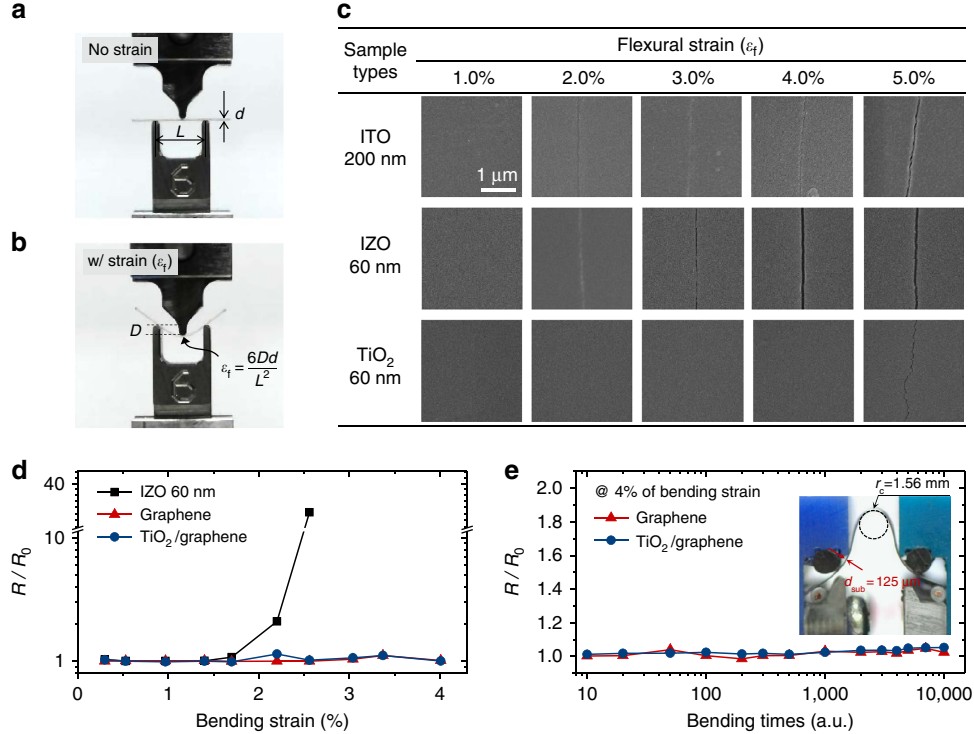

**Figure 5 | Flexural mechanical properties of TiO₂ films and electrical properties of TiO₂/Graphene under flexural strain.** (**a,b**) Images of the three-point bending-test set up (static loading) where flexural tensile strain ($\varepsilon_f$) is easily controlled via geometrical parameters shown in the pictures. (**c**) Scanning electron microscopy images for the top surfaces of 200-nm-thick ITO, 60-nm-thick IZO and 60 nm-thick TiO₂ on PET substrates after static loading at a specified $\varepsilon_f$. (**d**) The sheet resistance ($R$) of IZO (60 nm), graphene, TiO₂ (60 nm)/graphene on PET substrates as a function of flexural strain applied (measured after being bent 10 times at the specified strain (dynamic loading)). The sheet resistance is normalized to the initial value ($R_0$). (**e**) $R/R_0$ of graphene and TiO₂ (60 nm)/graphene on PET substrates in relation to the number of bending cycles at flexural strain of 4%.

common TiO₂ layer having the same thickness and can still lead to efficiency enhancement (see Supplementary Fig. 10, Supplementary Table 6 and Supplementary Note 3 for more discussion). Within this scheme, angular colour shift, typical for cavity-based OLEDs, can also be significantly suppressed for individual R, G, B as well as RGB-synthesized white with an only slight compromise in efficiency enhancement.

In summary, an ideal optical structure was proposed for graphene-based OLEDs, which leads to maximized efficiency while retaining their advantages in flexibility. The inherent thinness of graphene electrodes being considered, high-index TiO₂ layers were placed underneath graphene to enable cavity resonance enhancement. With the optical thickness of TiO₂ layers set at $\lambda/4$, the first-order $3\lambda/4$ cavity design was realized with the thickness of other layers falling within a range practical for electrical properties. The TiO₂ layer with the optical thickness of $\lambda/4$ played a role as a high-reflectance coating and thus improved the resonance enhancement. Conducting polymers of PEDOT:PSS or GraHIL adopted as hole-injection layers plays an additional yet substantial role as a low-index layer that further enhance the EQE by redistributing the optical power gained from reduction of SPP loss that limits the performance of first-order cavity OLEDs; and by increasing the reflectance from the bottom electrode assembly and thus leading to an additional improvement in Fabry–Perot resonance effect. With the proposed synergetic interplay between high- and low-index layers, the optimized OLEDs exhibited ultrahigh EQE and power efficiency of 40.8% and 160 lm W⁻¹; and 62.1% and 120 lm W⁻¹ for single- and multi-junction structures, respectively (64.7% and 250 lm W⁻¹; and 103% and 184 lm W⁻¹ with a half-ball lens attached), which are unprecedented for graphene-based OLEDs. Furthermore, efficient flexible OLEDs were demonstrated that

can be bent to a radius of 2.3 mm. Independent study revealed that the TiO₂ layer has an excellent flexural strain resistance unusual for most ceramic materials due to its crack deflection toughening mechanism. When combined with its high refractive index, this unique property enabled highly flexible OLEDs with significantly enhanced efficiency. Given the level of efficiency and bendability realized in this work, we believe the proposed method can pave the way for graphene-based OLEDs to become a next-generation light source balanced with both efficiency and form factor advantages.

## Methods

**Preparation of substrates, TiO₂ and graphene.** Polished glass substrates (Eagle XG, Corning) and those with the pre-coated ITO films ($<12\,\Omega\,\text{sq}^{-1}$; Shinan, Korea) were cleaned as reported earlier[34]. PET substrates (50 and 125 μm; SKC, Korea) were cleaned with IPA. Clear photoresist films of SU-8 (600 nm; Microchem) were spun at 2,000 r.p.m. for 30 s on PET substrates for planarization. SU-8 layers were baked at 100 °C for 10 min and cross-linked under ultra-violet light. On top of cleaned glass and planarized PET substrates, TiO₂ layers (GMEK, Korea) or IZO layers were deposited using a DC pulsed (500 W) or RF (120 W) sputtering process. Formation of MLG on a target substrate was done in one of the following two ways: one involved quadruple repetition of growth of a SLG on a copper foil and subsequent wet-transfer process to a target substrate (Method 1; for the devices in Fig. 3)[35,36]; the second way involved direct growth of MLG on a Ni layer coated on a SiO₂ (300 nm)/Si wafer held at 750 °C in an inductively coupled plasma system, followed by a wet transfer process (Method 2; for the devices in Fig. 4)[37]. In both cases, the wet transfer was assisted by spin-coated poly(methyl methacrylate). The MLG layers were p-doped in HNO₃ aqueous solution or by its vapour. The sheet resistance and transmittance (at $\lambda$ of 550 nm; normalized to that of a blank glass substrate) of doped MLG layers from Methods 1 and 2 were $92.5 \pm 9.4\,\Omega\,\text{sq}^{-1}$ and 90%, and $330.3 \pm 16.1\,\Omega\,\text{sq}^{-1}$ and 88%, respectively.

**OLED fabrication and evaluation.** GraHIL or PEDOT:PSS (mixed with IPA (3:1 by volume)) were spin-coated as a HIL on top of TiO₂ (55 nm)/MLG, MLG or ITO electrodes to result in 70 and 50-nm-thick films, respectively. The spin-coating

solution for GraHIL was prepared by mixing PEDOT:PSS (Clevios P VP AI4083) with PFI (Sigma-Aldrich) in 1:1 ratio[13]. The spin-coated HILs were then dried on a hotplate. The samples with spin-coated HIL were then loaded into a thermal evaporator for deposition of organic layers and metal electrodes of LiF (1 nm)/Al (>100 nm). The organic multilayer layer stack used for the devices in Fig. 3 (OS1) was TAPC (15 nm)/TCTA: Ir(ppy)₂acac (5 nm, 97:3 by volume)/CBP: Ir(ppy)₂ acac (5 nm, 96:4, by volume)/TPBi (55 nm), where TAPC, TCTA, CBP and TPBi refer to 4,4′-cyclohexylidenebis(N,N-bis(4-methylphenyl)benzenamine); Tris (4-carbazoyl-9-ylphenyl) amine; 4, 4′-Bis(N-carbazolyl)-1, 1′-biphenyl; and 2, 2, 2-(1, 3, 5-benzenetriyl)tris-(1-phenyl-1H-benzimidazole), respectively[13]. The multilayer layer stack used for the devices in Fig. 4 (OS2) was MoO₃ (5 nm)/CBP (20 nm)/1:1 co-host of CBP: B3PYMPM doped with 7% Ir(ppy)₂acac (20 nm)/ B3PYMPM (55 nm), where B3PYMPM refers to (bis-4, 6-(3, 5-di-3-pyridylphenyl)-2-methylpyrimidine). Note that, in the case of PEDOT:PSS-based devices, MoO₃ (Alfa Aesar) was deposited on top of PEDOT:PSS to ensure efficient hole injection.

Fabricated OLED devices were evaluated either in ambient air after encapsulation done in an inert atmosphere (the devices in Fig. 3) or as-prepared in an inert environment (the devices in Fig. 4). A computer-controlled and motorized goniometric system equipped with a calibrated photodiode (Thorlab, Inc.) and a fibre-optic spectrometer (StellarNet, Inc.) was used for full angle-resolved characterization of intensity and emission spectra. EQE was then estimated by taking a full account of the measured angle-dependent intensity and spectral variation. For devices with a half-ball lens (BK-7, 10 mm in diameter; Edmund Optics), the test was done with an integrating sphere. The lens was optically coupled on to the substrate using an index matching fluid (F-IMF-105, $n = 1.52$, Newport). Flexural strain measurement was done with a high-precision micromechanical test system (DTS Company, USA) according to ASTM standard (D790). The bending-cycle test of OLED devices and TiO₂/MLG on plastic substrates was done with a custom-made bending tester able to adjust the radius of curvature and the number of bending cycles.

**Optical analysis.** Optical modelling was done with a custom-made MATLAB code following the formalism summarized by Furno et al.[19] that takes a full account of waveguide and SPP modal excitation, Purcell factor and dipole-orientation effect. Optical constants of materials used for simulation were borrowed from the literature[38] or measured with spectroscopic ellipsometry, and are given in Supplementary Fig. 3. For simplicity, the organic multilayers (except for HILs) were assumed to be a single homogeneous organic layer with total thickness preserved, as organic layers used in this work have similar refractive indices in the visible spectral region as shown in Supplementary Fig. 3. For Ir(ppy)₂acac, anisotropic dipole orientation factor (a) of 0.23 ± 0.02, corresponding to ca. 77% horizontally oriented dipoles, the radiative quantum efficiency of 0.97 were used following the previous reports[39,40]. The dipole emitter was placed at a distance from the organic/cathode interface optimal for the maximum two-beam constructive interference. The spatial distribution of emitters, which may present in actual devices, was not taken into account in this study, as their optical influence was rather minor. The electron-hole balance efficiency was set at unity. Quantitative agreement was observed between calculation and experimental results in a fully angle- and spectrally resolved manner (Supplementary Fig. 11).

**Data availability.** The authors declare that the data supporting the findings of this study are available within the article and its Supplementary Information files. Numerical values of data shown as graphs are available upon request from the corresponding authors.

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

## Acknowledgements

This work was supported in part by the Basic Science Research Program through the National Research Foundation of Korea (NRF) funded by the Ministry of Science, ICT and Future Planning (MSIP) (CAFDC 4-1, 2007-0056090 (S.Y.); by the Center for

Advanced Soft-Electronics funded by the MSIP as Global Frontier Project (2014M3A6A5060947 (T.L.)), and by the Graphene Research Center Program of KAIST (S.Y. and S.C.). S.C. acknowledges grants from IT R&D Program of Ministry of Trade, Industry and Energy of Korea (MOTIE) (10044412).

## Author contributions

S.Y. and T.W.L. conceived an idea on graphene based OLEDs using $TiO_2$ layers and low-index HILs, and S.Y., T.W.L., S.Y.C. and T.S.K. designed associated experiments. J.L. and J.C. fabricated and tested rigid or flexible OLEDs with $TiO_2$/graphene/PED-OT:PSS. T.H.H. and H.K.S. fabricated and tested OLEDs with $TiO_2$/graphene/GraHIL. H.K.S, T.H.H and M.H.P. fabricated and tested the multi-junction OLED devices. D.Y.J. and H.K.S. carried out the growth and transfer of graphene and analysed their characteristics. H.S.C. made a custom MATLAB code for optical simulation, and H.S.C., E.K. and J.L. performed optical analysis. E.K. and J.L. prepared IZO samples. J.M.S. and T.S.K. designed and performed the 3-point bending test. All authors read and discussed the results of manuscript. J.L., T.H.H. and M.H.P. equally contributed as main authors. D.Y.J., J.M.S. and H.K.S. equally contributed as second authors. S.Y. and T.W.L. equally contributed as corresponding authors.

## Additional information

**Competing financial interests:** The authors declare no competing financial interests.

