## [Peer Review File · Nature Communications]

REVIEWERS' COMMENTS:

Reviewer #1 (Remarks to the Author):

I reviewed this manuscript some time ago for Nature Photonics. In the first round of review, I raised a few questions and it seems that the authors took my comments seriously and addressed those issues in detail in the revised version. I feel that in the current form it can be published in Nature Communications.

Reviewer #2 (Remarks to the Author):

I have been asked to re-review the manuscript by Seunghyup Yoo on the use of graphene electrodes and high index TiO₂ layers to improve the light extraction efficiency in OLEDs and at the same time provide a flexible electrode technology. The original manuscript submitted to Nature Photonics was already very impressive. However, back then I raised some concerns about the conceptual novelty of the work which in my view meant that the manuscript was maybe not fully suitable for publication in Nature Photonics.

However, even without the recent revisions I would have very strongly recommended the manuscript for publication in Nature Communications. (As I said in my previous report; this is a very strong demonstration. It combines a number of demanding state-of-the-art concepts in OLED technology in a highly original manner.)

In their rebuttal letter, the authors have now provided a very convincing discussion of my concerns regarding novelty. They also address the few further technical points, I had raised. This has also led to extensive revisions of the manuscript, which as a result of this I think has become even stronger.

I think this paper is ready for publication and I very strongly recommend that it is published in Nature Communications as soon as possible.

1. Response to Reviewers' comments

As shown below, all the reviewers recommended the present manuscript to be accepted 'as is' and thus did not indicate any further content revisions to make; therefore, we have focused on revising the manuscript according to the formatting guidelines kindly provided by Editor.